# Phylogenetic Analysis of Newcastle Disease Virus Isolated from Poultry in Live Bird Markets and Wild Waterfowl in Zambia

**DOI:** 10.3390/microorganisms12020354

**Published:** 2024-02-08

**Authors:** Annie Kalonda, Ngonda Saasa, Masahiro Kajihara, Naganori Nao, Ladslav Moonga, Joseph Ndebe, Akina Mori-Kajihara, Andrew Nalishuwa Mukubesa, Yoshihiro Sakoda, Hirofumi Sawa, Ayato Takada, Edgar Simulundu

**Affiliations:** 1Department of Biomedical Sciences, School of Health Sciences, University of Zambia, Lusaka 10101, Zambia; 2Department of Disease Control, School of Veterinary Medicine, University of Zambia, Lusaka 10101, Zambia; nsaasa@gmail.com (N.S.); j.ndebe@yahoo.com (J.N.); mukubesaandrew@gmail.com (A.N.M.); h-sawa@ivred.hokudai.ac.jp (H.S.); atakada@czc.hokudai.ac.jp (A.T.); 3Africa Centre of Excellence for Infectious Diseases of Humans and Animals, School of Veterinary Medicine, University of Zambia, Lusaka 10101, Zambia; 4Division of International Research Promotion, International Institute for Zoonosis Control, Hokkaido University, N20 W10, Kita-ku, Sapporo 001-0020, Japan; kajihara@czc.hokudai.ac.jp (M.K.); n-nao@czc.hokudai.ac.jp (N.N.); 5Hokudai Center for Zoonosis Control in Zambia, School of Veterinary Medicine, University of Zambia, Lusaka 10101, Zambia; sakoda@vetmed.hokudai.ac.jp; 6One Health Research Center, Hokkaido University, N18 W9, Kita-ku, Sapporo 001-0020, Japan; 7Department of Paraclinical Studies, School of Veterinary Medicine, University of Zambia, Lusaka 10101, Zambia; ladslavm@yahoo.com; 8Division of Global Epidemiology, International Institute for Zoonosis Control, Hokkaido University, N20 W10, Kita-ku, Sapporo 001-0020, Japan; akinam@czc.hokudai.ac.jp; 9International Collaboration Unit, International Institute for Zoonosis Control, Hokkaido University, N20 W10, Kita-ku, Sapporo 001-0020, Japan; 10Laboratory of Microbiology, Department of Disease Control, Faculty of Veterinary Medicine, Hokkaido University, N18 W9, Kita-ku, Sapporo 060-0818, Japan; 11Division of Biological Response Analysis, Institute for Vaccine Research and Development (IVReD), Hokkaido University, N21 W11, Kita-ku, Sapporo 001-0020, Japan; 12Division of Molecular Pathobiology, International Institute for Zoonosis Control, Hokkaido University, N20 W10, Kita-ku, Sapporo 001-0020, Japan; 13Global Virus Network, 725 W Lombard Street, Baltimore, MD 21201, USA; 14Macha Research Trust, Choma 20100, Zambia

**Keywords:** Newcastle disease virus, sub-genotype VII.2, phylogenetic analysis, live bird market, wild birds, poultry, Zambia

## Abstract

Poultry production is essential to the economy and livelihood of many rural Zambian households. However, the industry is threatened by infectious diseases, particularly Newcastle disease virus (NDV) infection. Therefore, this study employed next-generation sequencing to characterise six NDV isolates from poultry in Zambia’s live bird markets (LBMs) and wild waterfowl. Four NDV isolates were detected from 410 faecal samples collected from chickens in LBMs in Lusaka and two from 2851 wild birds from Lochinvar National Park. Phylogenetic analysis revealed that the four NDVs from LBM clustered in genotype VII and sub-genotype VII.2 were closely related to viruses previously isolated in Zambia and other Southern African countries, suggesting possible local and regional transboundary circulation of the virus. In contrast, the two isolates from wild birds belonged to class I viruses, genotype 1, and were closely related to isolates from Europe and Asia, suggesting the possible introduction of these viruses from Eurasia, likely through wild bird migration. The fusion gene cleavage site motif for all LBM-associated isolates was ^112^RRQKR|F^117^, indicating that the viruses are virulent, while the isolates from wild waterfowl had the typical ^112^ERQER|L^117^ avirulent motif. This study demonstrates the circulation of virulent NDV strains in LBMs and has, for the first time, characterised NDV from wild birds in Zambia. The study further provides the first whole genomes of NDV sub-genotype VII.2 and genotype 1 from Zambia and stresses the importance of surveillance and molecular analysis for monitoring the circulation of NDV genotypes and viral evolution.

## 1. Introduction

Newcastle disease (ND) is a highly contagious and one of the most devastating avian viral diseases affecting the global poultry industry and wild birds [1]. It is classified as a notifiable disease by the World Organization for Animal Health (WOAH) in its highly pathogenic form [2]. Outbreaks of the disease result in significant economic losses associated with high morbidity and mortality rates of approximately 100% in severe cases, especially in naïve or poorly vaccinated chickens, and drops in egg production in well-vaccinated layers [3]. Whilst ND is globally distributed, developing countries are severely impacted as proper control measures against the disease are lacking [4]. 

ND is caused by virulent strains of avian paramyxovirus 1 (APMV-1), also known as Newcastle disease virus (NDV, used hereafter), belonging to the species *Orthoavulavirus javaense* of the genus *Orthoavulavirus* of the family *Paramyxoviridae* in the order *Mononegavirales* [5,6]. NDV is an enveloped, single-stranded, negative-sense RNA virus with a non-segmented genome ranging from 15,186 to 15,198 kb in size. Its genome has six open reading frames (ORF), which encode six major structural proteins, namely, nucleoprotein (NP), phosphoprotein (P), matrix protein (M), fusion protein (F), hemagglutinin-neuraminidase (HN), and the RNA-dependent RNA polymerase (L) in the order 3′-NP-P-M- F-HN-L-5′, and two non-structural proteins (V and W) generated by the P-gene mRNA editing [7,8]. The F and HN proteins are the main virulence factors of the virus [9,10] and play essential roles in the assembly and development of enveloped viruses, determining tropism in the host and tissues and facilitating the attachment of viruses to the host cell [11]. 

Based on pathogenicity in chickens, NDVs are classified into three major pathotypes: velogenic (highly virulent), mesogenic (moderately virulent), and lentogenic (avirulent or low virulent). Velogenic strains are further classified into viscerotropic and neurotropic velogenic strains and are responsible for severe disease and cause significant mortality in susceptible flocks [4,12]. According to the WOAH, virulent NDV strains, which include both mesogenic and velogenic strains, must meet one of the following criteria: (i) have an intracerebral pathogenicity index (ICPI) in day-old chicks (*Gallus gallus*) of 0.7 or greater; or (ii) have multiple basic amino acids at the C terminus of the F2 protein and phenylalanine at residue 117, which is the N terminus of the F1 protein [13]. The virulent and avirulent NDV strains have the sequence of ^112^R/K–R–Q/K/R–K/R–R–F^117^ and ^112^G/E–K/R–Q–G/E–R–L^117^ in the F protein cleavage site, respectively [13].

NDV strains have also been classified into two distinct classes based on the phylogenetic analysis of the full-length F gene, namely class I and class II, all within a single serotype. Class I viruses belong to a single genotype (genotype I), consisting primarily of avirulent NDV strains, usually from wild birds, that occasionally spill over into poultry. In contrast, the more genetically diverse class II viruses are currently divided into 21 genotypes (I–XXI) with several sub-genotypes, including avirulent and virulent isolates from a wide range of domestic and wild birds [5]. Class II viruses have been known to have caused at least four ND panzootics, and viruses from genotype VII are responsible for the fourth panzootic [14], which originated in Southeast Asia, with the earliest known outbreaks beginning around 1985 and is believed to be on-going. In addition, genotype VII, comprising virulent NDV strains, is the most prevalent genotype circulating worldwide and is associated with many outbreaks among chickens. Moreover, non-commercial poultry and wild waterfowl seem to be the natural reservoirs for both virulent and avirulent NDVs. In Africa, studies have reported the isolation of both virulent and avirulent strains of NDV from sick and healthy poultry. Viruses from genotypes I–VIII, XI, XIII-XIV, and XVII–XVIII have been detected on the continent [15].

In Zambia, NDV was first detected in 1952 in the Southern Province, where 15 outbreaks were recorded in that year, although the pathotypes of the isolates remained unknown [16]. By the early 1980s, velogenic NDV strains had been detected in different parts of the country, and ND was considered enzootic throughout the country [16]. Since then, the government enforced a vaccination policy in commercial poultry farms. However, ND continues to contribute to a high burden of morbidity and mortality among village chickens nationwide, with a reported seroprevalence of 36.5% and 73.9% in 1994 and 2012, respectively [17,18]. Moreover, ND is considered the leading cause of death in chickens in Zambia [19], negatively affecting the livelihoods of poor rural households who depend on poultry as a source of protein and income. Recent genetic studies conducted in Zambia revealed the circulation of NDVs from genotypes VII.2 (previously VIIh) and XIII among chickens in the Eastern Province [20,21]. Despite these studies, which were very limited in geographical coverage, there is a paucity of data regarding the circulation of NDV in live bird markets (LBMs) and wild waterfowls. The present study describes the phylogenetic analysis of NDV isolated from wild waterfowl and poultry during avian influenza virus (AIV) surveillance activities in selected parts of Zambia. 

## 2. Materials and Methods

### 2.1. Sampling Sites and Sample Collection

The study samples were collected in 2015, 2020, and 2021 from wild waterfowl and poultry in Zambia (Table 1) as part of the AIV surveillance activities. Results for AIV surveillance in wild waterfowl have been published elsewhere [22]. A total of 2851 fresh faecal samples were collected from the wild birds in Lochinvar National Park (LNP), a bird sanctuary that is home to migratory birds. On the other hand, 1150 samples were collected from exotic and indigenous poultry from 21 LBMs from 11 towns across three provinces in Zambia (Figure 1). The sampling sites were purposively chosen based on their high population density of wild birds and the availability of LBMs. Moreover, the major poultry markets and poultry production are concentrated along the rail line from Livingstone in the Southern Province of Zambia to Kasumbalesa on the Zambia-Democratic Republic of the Congo border in the Copperbelt Province (Figure 1). Traders in the LBMs acquire their bird stocks from various commercial and backyard poultry farms across the country, and the consumers buy the birds for restocking [especially indigenous (village/traditional) chicken keepers] and the home and hospitality businesses for slaughter. Typically, poultry in LBMs are kept in wire-mesh cages with different bird species mixed in the same cage (Appendix A). Faecal sample collection, transportation, and storage were performed as previously described [23]. Fresh faecal droppings were collected, placed in tubes, and transported in cool boxes containing ice packs to the University of Zambia, School of Veterinary Medicine Virology laboratory, where they were stored at −80 °C before processing within 72 h.

### 2.2. Virus Isolation and Identification

Faecal samples were processed as per the previously described protocol [13,22]. About 0.5 mL of supernatant fluid from each sample was inoculated in duplicate into the allantoic cavity of 9 to 11-day-old embryonated chicken eggs. Inoculated eggs were incubated at 37 °C for 48 h and then chilled overnight at 4 °C. Allantoic fluid was harvested three days post-inoculation and subjected to haemagglutination (HA) assay using 0.5% chicken erythrocytes. All HA-positive samples were subsequently subjected to a haemagglutination-inhibition (HI) test with NDV-specific antiserum according to the standard procedure [13].

### 2.3. Sequencing and Genetic Analysis

Sequencing was performed as previously described [22]. Viral RNA was extracted from the HA-positive and HI-confirmed allantoic fluids using the QIAamp Viral RNA Mini kit (Qiagen, Hilden, Germany) according to the manufacturer’s protocol. The extracted RNA was subjected to next-generation sequencing using the Illumina MiSeq System (Illumina, San Diego, CA, USA). MiSeq libraries were prepared using the KAPA RNA Hyper Prep Kit and sequenced using the MiSeq Reagent Kit v3 (600 cycles) (Illumina, San Diego, CA, USA). Sequence reads were mapped to reference sequences of NDV, and the consensus sequence was rebuilt until all mismatches were resolved using the CLC Genomic Workbench, version 22.0 (CLC bio, Aarhus, Denmark). All viral sequences were submitted to GenBank, accession numbers OR964056–OR964061.

For phylogenetic analysis, a dataset was prepared using the pilot datasets of the full-length NDV F gene sequences proposed by Dimitrov et al. [5], in which the Zambian isolates from the current study were inserted. Additionally, full-length NDV F gene reference sequences with the highest sequence similarity to the sequences obtained in this study as determined by the Basic Local Alignment Search Tool (BLAST) were downloaded from the National Centre for Biotechnology Information (NCBI) database and included in the dataset. The dataset also included previously sequenced Zambian NDVs and those from neighbouring countries. Multiple sequence alignment was performed using the Multiple Alignment with Fast Fourier Transformation (MAFFT) (https://mafft.cbrc.jp/alignment/software/) (accessed on 1 December 2022) according to default parameters [24]. The aligned sequences were manually edited and trimmed using Geneious Prime^®^ v2022.2.2. The maximum likelihood method was applied to construct the phylogenetic tree using the Tamura–Nei evolutionary model [24], available in Molecular Evolutionary Genetics Analysis version X (MEGA X) software [25,26] with 1000 bootstrap replicates.

The per cent similarities of the Zambian and other NDV isolates were assessed using BLAST, and amino acid (AA) sequence comparison of the F protein was conducted using GENETYX Version 12.0 (GENETYX Corp., Tokyo, Japan).

### 2.4. Ethics

Permission to conduct the study in LNP and LBMs was obtained from the Ministry of Tourism and Arts and the Ministry of Livestock and Fisheries in Zambia, respectively. The study was approved by the University of Zambia Biomedical Research Ethics Committee (reference number 616-2019).

## 3. Results

### 3.1. Virus Isolation and Identification

A total of six NDV isolates were detected from the LBMs and LNP in Zambia. We found four samples positive for NDV among the 1150 samples collected from domestic birds in LBMs (0.3% positivity rate) and two positive samples among 2851 samples collected from wild birds in the LNP (0.07% positivity rate). All four NDV isolates from poultry were obtained from apparently healthy chickens at one of the major and largest LBM in Lusaka, the capital city of Zambia. Two isolates were obtained from indigenous chickens, while the other two were from broiler chickens. Two isolates from wild waterfowl were obtained from a duck and an ibis. No NDVs were isolated from LBMs in Copperbelt and Southern provinces. 

### 3.2. Genomic Characteristics of Newcastle Disease Virus

The genome lengths of the isolates described in this study followed the ‘rule-of-six’, which is considered an essential feature for efficient replication of paramyxoviruses [27], and they all had the genome organisation 3′-N-P-M-F-HN-L-5′, as has been observed for all known NDV isolates. Sequence analysis showed that out of the six Zambian NDV isolates, four had either 9–14 nucleotides missing at either the 5′ or 3′ ends, giving a consensus genomic sequence length ranging from 15,178 to 15,183 nucleotides (nt). The other two isolates, NDV/duck/Zambia/708/2021 and NDV/Chicken/Zambia/48/2020 had sequence lengths of 15,166 and 15,122 nt, respectively, due to missing sequences at the 5′ or 3′ ends. The nucleotide per cent identity of the NDV genome from the four poultry isolates was 99.9–100%, while that between the two isolates from the wild waterfowl was 98.14%. The full-length F gene sequence similarities between the Zambian NDV isolates and other strains were also assessed using BLAST search, and the identity similarity with the top ten hit strains was 98.9–97.4% (Appendix A). 

### 3.3. Phylogenetic Analysis of Newcastle Disease Virus

Phylogenetic analysis of the four NDV isolates from poultry using the full-length F gene revealed that all the isolates clustered together and were closely related to NDV detected in the Eastern Province of Zambia in 2015. These viruses belonged to class II viruses, genotype VII, sub-genotype VII.2 (Figure 2). Furthermore, the analysis also indicated that the NDV isolates obtained from wild birds clustered with strains from Eurasia among class I viruses, genotype 1 (Figure 2). 

### 3.4. Genetic Analysis of the F Gene Cleavage Site

Analysis of the amino acid (AA) proteolytic cleavage site of the F gene revealed that the F0 precursor cleavage site of the four Zambian isolates from poultry contained multiple basic amino acid residues at the C-terminus of the F2 protein and a phenylalanine (F) residue at the N-terminus of the F1 protein (^112^RRQKR|F^117^) which is a characteristic feature of mesogenic and velogenic strains of NDV. In comparison, the two isolates from wild birds possessed the motif ^112^ERQER|L^117^ of the lentogenic strains of NDV, as shown in Table 2.

## 4. Discussion

This study is the first to report the isolation of NDV from domestic and wild birds in Zambia from a LBM and LNP, respectively. The viruses were isolated as a result of risk-based active surveillance for avian influenza viruses, highlighting the importance of active surveillance of these viruses. This is the first time such surveillance has been implemented in domestic birds in LBMs in Zambia. Importantly, no avian influenza viruses were detected in domestic birds in LBMs in this study. Previous studies have reported the isolation of NDV from backyard poultry in the Eastern Province of Zambia [20,21]. In addition, most work on NDV in many parts of the world has focused on poultry, where both virulent and avirulent strains of class II have been reported, with limited studies conducted in wild birds [3,28]. 

The present study demonstrated the isolation of NDV in broiler and village chickens sold at a LBM in Lusaka and wild birds in the LNP, with a positivity rate of 0.3% and 0.07%, respectively. The positivity rate of NDV recorded in chickens at LBMs in the current study was lower than 4.36–91.67% reported in Ghana and Tanzania [29], 65.9% in Egypt [30], 54% in Nigeria [31], and 45% in Libya [32]. Moreover, the positivity rate of NDV in wild birds was also lower compared to 2–10% reported in other African countries [28,33,34,35]. The observed difference in the positivity rates could be attributed to the differences in the health status of the birds sampled, the type of samples collected, and the methods of detection used. For example, other studies collected organ samples and cloacal or oropharyngeal swabs from sick or both sick and apparently healthy birds and used RT-PCR for detection [31,32], which may increase the detection rate. The low isolation rate of NDV in poultry could indicate some success in containing the disease through ND vaccinations in the country. However, this study could not attribute the low isolation rate to vaccination because no vaccination status of poultry was collected during sampling. In addition, NDVs from poultry were all isolated in the summer months of the dry season, which is not surprising as it has been previously reported that ND epidemics usually occur in the hot, dry season [18,36], possibly due to climatic stress, leading to seasonal occurrence [37]. The detection of NDV in wild birds is very cardinal since wild birds are known to be natural reservoirs of NDV of low virulence, which can spill over into poultry. 

Despite the low positivity rate, isolation of NDV from poultry sold at LBMs is very cardinal because LBMs may likely act as foci of uninterrupted replication, sustained maintenance, and dissemination of NDV variants. Furthermore, LBMs are places where birds mix in unhygienic and crowded cages, usually with no biosecurity measures, making the transmission of any avian disease possible. Interestingly, all positive samples from apparently healthy birds were detected from one LBM in Lusaka District in Lusaka Province, commonly known as Soweto Market. Soweto LBM is the largest market in Lusaka City, and birds sold at this market originate from various parts of the country. Moreover, the market serves as the chicken source for many urban markets within the country and for farmers intending to rear chickens, especially backyard poultry farmers. Therefore, LBMs such as Soweto can foster the transmission and dissemination of NDV to other areas. Furthermore, the four NDV isolates obtained from poultry were found to be genetically identical based on nucleotide per cent identity, suggesting a possible occurrence of a localised outbreak within the Soweto LBM. Although no NDV was detected in other LBMs surveyed, it is imperative to continue monitoring this important pathogen in these and other LBMs. 

Phylogenetic analysis revealed that the NDV isolates obtained from chickens belonged to class II, genotype VII, sub-genotype VII.2 previously classified as sub-genotype VIIh [5,38]. Viruses from genotype VII are prevalent worldwide and are responsible for the most recent ND panzootic. Detection of genotype VII, sub-genotype VII.2 (VIIh) isolates has been reported previously in Botswana, Malawi, Mozambique, South Africa, Zambia, and Zimbabwe [21,39,40]. The circulation of this endemic NDV strain in Southern Africa could be attributed to the uncontrolled cross-border trading of live birds [21]. Although genotype XIII was detected in the Eastern province of Zambia in a previous study [20], it was not detected in the current study, probably due to geographical differences as well as differences in the poultry targeted. The previous study targeted indigenous backyard poultry, and genotype XIII virus was isolated from clinically sick chickens [20]. The isolates in the current study were closely related to previous Zambian NDVs and those detected in neighbouring countries in Southern Africa. The close relationship between the 2015 Zambian NDV isolates and those from the current study may indicate that this NDV variant may be endemic in the country. Furthermore, the detection of class II genotype VII viruses emphasises the need for continued molecular surveillance of NDV to monitor the evolution and distribution of endemic strains in Zambia and the African continent. 

The findings further demonstrated that the two NDV isolates from wild waterfowl belonged to class I, genotype 1 strains and were closely related to NDV strains from Europe and Asia. The close relationship of our isolates with those obtained from wild birds in Eurasia may implicate wild bird migration as a conduit for the introduction of these viruses into the country. Therefore, genotype 1 NDV isolates from the current study could have been introduced into the wild bird population in Zambia by Eurasian migratory birds as the isolation period corresponds to the season when migratory birds are resident in the Zambian wetlands. In Zambia, Eurasian migratory birds begin to arrive in November until April. Studies in Africa have reported NDVs belonging to classes I and II in wild waterfowl. For instance, Snoeck et al. [41] reported class I (avirulent) and class II, genotype XVIII (virulent) NDV strains in wild birds in Nigeria and Côte d’Ivoire, respectively. Other NDV genotypes reported in wild birds in Africa include avirulent class II, genotype II and virulent genotype VII NDV strains [33,34,35]. Thus, molecular surveillance in world birds is critical to designing control measures for NDV as previous studies suggest that wild birds may not only act as reservoirs of low virulence strains but may also play a crucial role in the epidemiology of different variants of NDV persisting in Africa, including virulent strains responsible for outbreaks in poultry [42,43]. 

In the present study, the four isolates obtained from chickens were classified as virulent based on the amino acid sequence motif ^112^RRQKR|F^117^ at the F protein cleavage site. In contrast, the two NDVs from wild birds were classified as avirulent as they possessed amino acid sequence motif ^112^ERQER|L^117^, which is typical of avirulent strains. Moreover, several studies have also reported the isolation of virulent strains of NDV in chickens [44,45,46] and avirulent strains in wild birds [33,46,47]. While this study reported avirulent variants of NDV in wild birds, virulent class II variants responsible for many outbreaks have also been isolated in wild birds [3,48], indicating the importance of these birds in disseminating the virus.

## 5. Conclusions

The study demonstrated for the first time the circulation of genotype VII.2 and genotype 1 NDV in poultry in LBMs and wild waterfowl, respectively, in Zambia. Phylogenetic analyses suggest the endemicity of genotype VII.2 viruses in Zambian poultry. Additionally, the genotype 1 isolates were closely related to viruses isolated in Europe and Asia, suggesting the possible introduction of these viruses into the Zambian ecosystem by Eurasian migratory birds. Given the economic importance of the poultry industry, the present findings stress the necessity of applying more strict biosecurity measures and management practices at LBM and applying measures that limit contact between wild birds and poultry flocks to curtail possible NDV spillover events. 

## Figures and Tables

**Figure 1 microorganisms-12-00354-f001:**
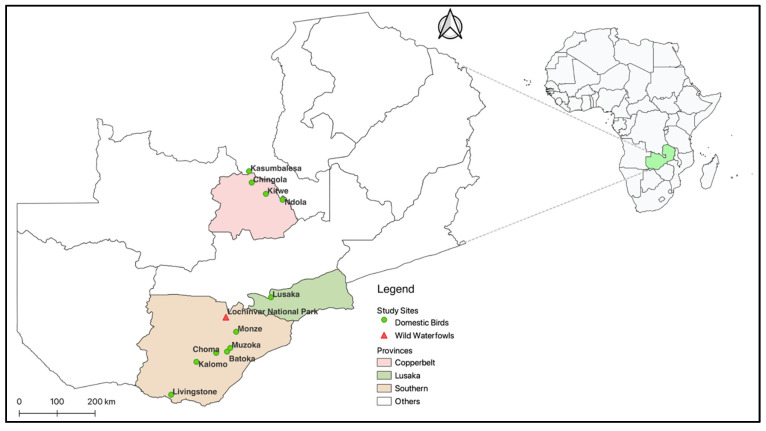
Sampling sites in three provinces of Zambia. Domestic birds were sampled in live bird markets. The maps were generated using Quantum Geographic Information System (QGIS) version 3.10 (http://www.qgis.org) accessed on 30 August 2023.

**Figure 2 microorganisms-12-00354-f002:**
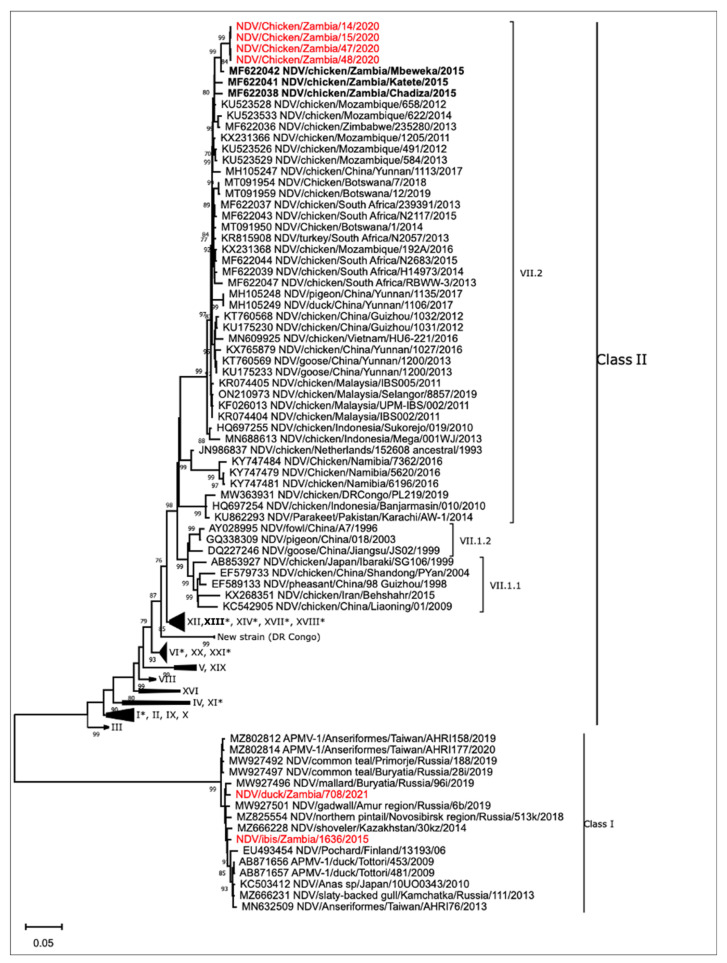
Phylogenetic tree of class I and II NDVs isolated in Zambia based on the full-length nucleotide sequence of the fusion gene. The sequences from this study are shown in red text, while those previously isolated in Zambia are in bold. Genotypes identified in Africa are indicated with an asterisk. The Roman numerals represent the genotypic classification of the viruses using the current nomenclature by Dimitrov and colleagues [5].

**Table 1 microorganisms-12-00354-t001:** Characteristics of collected samples.

Characteristics	Variable	No. of Samples Collected
**Poultry/Domestic Birds**	Sampling Sites	
	Copperbelt Province	402
	Lusaka Province	410
	Southern Province	338
	Bird Species	
	Chickens	1077
	Duck	24
	Guinea Fowl	19
	Quails	13
	Goose	8
	Turkey	5
	Dove	2
	Ostriches	2
	Year of Collection	
	2020	812
	2021	338
**Wild Birds**	Sampling Site	
	Southern Province—Locinvar ^1^	2851
	Bird Species	
	Ducks	1214
	Geese	1227
	Ibises	302
	Pelicans	98
	White Egrets	10
	Year of Collection	
	2015	1921
	2020	242
	2021	688

^1^ Lochinvar National Park.

**Table 2 microorganisms-12-00354-t002:** Characteristics of Zambian Newcastle disease virus isolates analysed in this study.

Virus Strain	Date of Isolation	Genotype	F Protein Cleavage Site
NDV/chicken/Zambia/14/2020	October 2020	VII.2	^112^RRQKR|F^117^
NDV/chicken/Zambia/15/2020	October 2020	VII.2	^112^RRQKR|F^117^
NDV/chicken/Zambia/47/2020	October 2020	VII.2	^112^RRQKR|F^117^
NDV/chicken/Zambia/48/2020	October 2020	VII.2	^112^RRQKR|F^117^
NDV/ibis/Zambia/1636/2015	November 2015	1	^112^ERQER|L^117^
NDV/duck/Zambia/708/2021	December 2021	1	^112^ERQER|L^117^

## Data Availability

The data presented in this study are available in this article and Appendix A.

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
