# Peer review of "Phylogenetic Analysis of Newcastle Disease Virus Isolated from Poultry in Live Bird Markets and Wild Waterfowl in Zambia"

_microorganisms, 2024, doi:10.3390/microorganisms12020354_

Round 1

Reviewer 1 Report

Comments and Suggestions for Authors

To Editor, Authors

No doubt, the manuscript “Phylogenetic Analysis of Newcastle Disease Virus Isolated from Poultry in Live Bird Markets and Wild Waterfowl in Zambia” is of interest for virologists. And also for molecular epidemiologists of poultry viruses and veterinarian authorities. Authors analyzed the NDV viruses in Zambia for the first time.

Actually data obtained needs to be considered when deciding on prevention and control. Introduction and aims are clear and logic.

Results and Discussion are interesting and have significant value for NDV virus researchers

But still has some questions to be addressed.

Comments:

1.      Have you identified the species of wild ducks? what species of wild duck has one isolate been isolated from?

2.      It would be good to give more details on vaccination in Zambia. is there a chance that there are vaccinated poultry in live poultry markets? It is unclear. line 277: you indicate the possible reasons for the low percentage of the virus. Could vaccination be one of them if it was carried out? Line 282: information on vaccination status was not collected during sampling. however, is there any information from the province at all? is there a vaccination of a private farmstead?

3.      The details of the isolation of four viruses on the market in Lusaka are interesting. According to the numbers in the name of the strain, they were isolated in the same place and at the same time. Is it so?

If the authors had also provided a comparison of the genome identity of these four strains, it would have helped answer the question: is this one strain that caused the local outbreak in certain market? You could detect just one outbreak (with isolation 4 same isolates) out of 1150 total samples from different places. All of the above may mean that monitoring has not detected NDV, except for one local outbreak in one market in Lusaka. This is for discussion.

4.      In the discussion, I found that 2 birds came to the market from the village, and the others were broilers - i.e. from different places/sources. if all four of them are the same virus, then the market itself may be the source of infection.

5.      Since this study is part of the avian influenza program, the interesting question is whether influenza viruses were isolated from domestic birds at that time? This is the first time such surveillance has been implemented in domestic birds in LBMs in Zambia. at least could be a mention or a link to the results.

6.      Line 70 – It may be good to give the genome size for different variants of the virus and in the format from ... to... kb.

7.      Line 141 – It would be possible to add a link to the manual for collecting, storing and transporting samples.

8.      3.3 Phylogenetic Analysis of Newcastle Disease Virus – Maybe it makes sense to make a tree of genome-wide sequences? Fusion is used to determine the evolutionary relationships between different isolates; fusion is used to determine virulence; The ready-made Dimitrov database (2019) is used for rapid typing of detected isolates. But it's not just fusion that affects virulence. When assessing risks and predicting the impact on poultry farming in the country, it is necessary to take into account changes in the whole genome. Instead of a tree, you can describe amino acid substitutions relative to the nearest strain with the described biological properties (if any).

It is just suggestion to the authors

9.      Line 284 – It is indicated that the highest percentage of virus release in poultry is observed in the summer. This proposal also indicates that the authors' results are consistent with the published reports. But the entire paragraph is devoted to the author's assumptions, explaining why the percentage of virus isolation in their study is significantly less than published studies. Based on this, I do not understand why line 284 says that the authors' results are consistent with those given. In the current study, the percentage of virus release in the summer was 0.3% and 0.07%. In the given links in the same period – 53% and 56%.

Author Response

Reviewer 1

No doubt, the manuscript “Phylogenetic Analysis of Newcastle Disease Virus Isolated from Poultry in Live Bird Markets and Wild Waterfowl in Zambia” is of interest for virologists. And also, for molecular epidemiologists of poultry viruses and veterinarian authorities. Authors analyzed the NDV viruses in Zambia for the first time. Actually, data obtained needs to be considered when deciding on prevention and control. Introduction and aims are clear and logic.

Results and Discussion are interesting and have significant value for NDV virus researchers

But still has some questions to be addressed.

Comments:

  1. Have you identified the species of wild ducks? what species of wild duck has one isolate been isolated from?

Response: No, we did not identify (confirm) the species of wild ducks using molecular methods. The ducks were identified morphologically, hence, we tried to be cautious by not specifying the bird species because when sampling we saw various duck species congregating together and these included white-faced whistling ducks, yellow-billed ducks and others. Hence, it was safer to refer to them as “wild duck” in this manuscript.

  1. It would be good to give more details on vaccination in Zambia. is there a chance that there are vaccinated poultry in live poultry markets? It is unclear. line 277: you indicate the possible reasons for the low percentage of the virus. Could vaccination be one of them if it was carried out? Line 282: information on vaccination status was not collected during sampling. however, is there any information from the province at all? is there a vaccination of a private farmstead?

Response: In Zambia, the poultry system is based on two systems that are commercial and backyard poultry. The commercial is where broilers or layers are obtained from hatcheries and reared on commercial feed and in properly designed chicken houses. Strict vaccination schedules for important avian diseases such as ND are followed, and the lentogenic LaSota/46 vaccine strain is widely used. Hence, most of the layers and broiler chickens sold in the markets are vaccinated. The second system is the village or backyard production system, is where chickens scavenge for food and there is no fixed schedule for vaccinations. However, in the recent past, the government through the veterinary services office and non-governmental organisations embarked on ND Vaccine sensitization campaigns, especially in ND outbreak-prone areas, which has seen backyard poultry/village chickens being vaccinated. Therefore, there higher chance that some or most of the chickens in the live bird markets might have been vaccinated. The sentence in the manuscript has been changed to “The low isolation rate of NDV in poultry could indicate some success in containing the disease through ND vaccinations in the country. However, this study could not attribute the low isolation rate to vaccination because no vaccination status of poultry was collected during sampling”

  1. The details of the isolation of four viruses on the market in Lusaka are interesting. According to the numbers in the name of the strain, they were isolated in the same place and at the same time. Is it so? If the authors had also provided a comparison of the genome identity of these four strains, it would have helped answer the question: is this one strain that caused the local outbreak in certain market? You could detect just one outbreak (with isolation 4 same isolates) out of 1150 total samples from different places. All of the above may mean that monitoring has not detected NDV, except for one local outbreak in one market in Lusaka. This is for discussion.

Response: Thank you for your suggestions: Yes, they were isolated in the same market in Lusaka on the same day.  The nucleotide percent identity of the NDV isolates from this study has been added to the manuscript in lines 236-237 and reads “The nucleotide percent identity of the NDV genome from the four poultry isolates was 99.9 – 100% while that between the two isolates from the wild waterfowl was 98.14%”. This point has also been discussed in lines 331–334 and reads “Furthermore, the four NDV isolates obtained from poultry were found to be genetically identical based on nucleotide percent identity, suggesting the possible occurrence of a localized outbreak within the Soweto LBM. Although no NDV was detected in other LBMs surveyed, it is imperative to continue monitoring this important pathogen in these and other LBMs”.

  1. In the discussion, I found that 2 birds came to the market from the village, and the others were broilers - i.e., from different places/sources. if all four of them are the same virus, then the market itself may be the source of infection.

Response: This study involved broiler chickens (exotic) and indigenous chickens (village chickens). The point of the market being the possible source of infection has been discussed in lines 303–334.

  1. Since this study is part of the avian influenza program, the interesting question is whether influenza viruses were isolated from domestic birds at that time? This is the first time such surveillance has been implemented in domestic birds in LBMs in Zambia. at least could be a mention or a link to the results.

Response: The results for AIVs in LBMs have been included in lines 277–279 and the sentence now reads “This is the first time such surveillance has been implemented in domestic birds in LBMs in Zambia. Importantly, no avian influenza viruses were detected in domestic birds in LBM in this study”.

  1. Line 70 – It may be good to give the genome size for different variants of the virus and in the format from ... to... kb.

Response: The sentence has been modified to read “NDV is an enveloped, single-stranded, negative-sense RNA virus with a non-segmented genome ranging from 15,186 – 15,198 kb in size”.

  1. Line 141 – It would be possible to add a link to the manual for collecting, storing and transporting samples.

Response: A sentence with citation has been added and reads “Faecal sample collection, transportation and storage was done as previously described (WHO, 2002)”.

  1. Phylogenetic Analysis of Newcastle Disease Virus – Maybe it makes sense to make a tree of genome-wide sequences? Fusion is used to determine the evolutionary relationships between different isolates; fusion is used to determine virulence; The ready-made Dimitrov database (2019) is used for rapid typing of detected isolates. But it's not just fusion that affects virulence. When assessing risks and predicting the impact on poultry farming in the country, it is necessary to take into account changes in the whole genome. Instead of a tree, you can describe amino acid substitutions relative to the nearest strain with the described biological properties (if any). It is just suggestion to the authors

Response: Thank you very much for your valuable suggestion. However, for this manuscript, we prefer to leave the analysis as presented.

  1. Line 284 – It is indicated that the highest percentage of virus release in poultry is observed in the summer. This proposal also indicates that the authors' results are consistent with the published reports. But the entire paragraph is devoted to the author's assumptions, explaining why the percentage of virus isolation in their study is significantly less than published studies. Based on this, I do not understand why line 284 says that the authors' results are consistent with those given. In the current study, the percentage of virus release in the summer was 0.3% and 0.07%. In the given links in the same period – 53% and 56%.

Response: The sentence and citation has been removed and the sentence now reads “In addition, NDV from poultry were all isolated in the summer months of the dry season which is not surprising as it has been previously reported that ND epidemics usually occur in the hot dry season [18, 37] possibly due to climatic stress, leading to seasonal occurrence [38].”

Reviewer 2 Report

Comments and Suggestions for Authors

Newcastle disease is considered the leading cause of death in chickens in developing countries, including Zambia. This disease negatively affects the livelihoods of poor rural households who depend on the poultry sector as a source of protein and income.

Currently, there is a lack of data regarding the circulation of NDV in live bird markets and wild waterfowls. Therefore, the authors aimed to describe the genetic relationship of Newcastle disease virus variants isolated from wild waterfowl and poultry during avian influenza virus surveillance activities in selected parts of Zambia.

The study is commendable, employing an exemplary methodology that incorporates next-generation sequencing. The main text, tables, and figures are clear, revealing for the first time the circulation of genotype VII.2 and genotype 1 variants of Newcastle disease virus in poultry in live bird markets and wild waterfowl, respectively, in Zambia.

I only have a minor comment regarding the use of the word "strain". I suggest replacing it with "variant" throughout the text.

Comments on the Quality of English Language

NA

Author Response

Reviewer 2

Newcastle disease is considered the leading cause of death in chickens in developing countries, including Zambia. This disease negatively affects the livelihoods of poor rural households who depend on the poultry sector as a source of protein and income. Currently, there is a lack of data regarding the circulation of NDV in live bird markets and wild waterfowls. Therefore, the authors aimed to describe the genetic relationship of Newcastle disease virus variants isolated from wild waterfowl and poultry during avian influenza virus surveillance activities in selected parts of Zambia. The study is commendable, employing an exemplary methodology that incorporates next-generation sequencing. The main text, tables, and figures are clear, revealing for the first time the circulation of genotype VII.2 and genotype 1 variants of Newcastle disease virus in poultry in live bird markets and wild waterfowl, respectively, in Zambia.

  1. I only have a minor comment regarding the use of the word "strain". I suggest replacing it with "variant" throughout the text.

Response: Thank you for your comment. The word “strain” has been replaced with “isolates” for NDV obtained in this study and variants when referring to NDV isolated in other studies. However, the word strain has been retained in places where the word fits the description such as the classification of NDV according to pathotypes such as “velogenic strains, lentogenic strains among others”.